# Surface-Functionalized Nanoparticles as Efficient Tools in Targeted Therapy of Pregnancy Complications

**DOI:** 10.3390/ijms20153642

**Published:** 2019-07-25

**Authors:** Baozhen Zhang, Ruijing Liang, Mingbin Zheng, Lintao Cai, Xiujun Fan

**Affiliations:** 1Center for Reproduction and Health Development, Institute of Biomedicine and Biotechnology, Shenzhen Institutes of Advanced Technology, Chinese Academy of Sciences, Shenzhen 518055, China; 2Guangdong Key Laboratory of Nanomedicine, CAS Key Lab for Health Informatics, Institute of Biomedicine and Biotechnology, Shenzhen Institutes of Advanced Technology, Chinese Academy of Sciences, Shenzhen 518055, China; 3University of Chinese Academy of Sciences, Beijing, 100049, China; 4Dongguan Key Laboratory of Drug Design and Formulation Technology, Key Laboratory for Nanomedicine, Guangdong Medical University, Dongguan 523808, China

**Keywords:** surface-functionalized nanoparticles, pregnancy complications, placenta target, uterus target, placenta-specific exosomes, plCSA-BP, transplacental transport

## Abstract

Minimizing exposure of the fetus to medication and reducing adverse off-target effects in the mother are the primary challenges in developing novel drugs to treat pregnancy complications. Nanomedicine has introduced opportunities for the development of novel platforms enabling targeted delivery of drugs in pregnancy. This review sets out to discuss the advances and potential of surface-functionalized nanoparticles in the targeted therapy of pregnancy complications. We first describe the human placental anatomy, which is fundamental for developing placenta-targeted therapy, and then we review current knowledge of nanoparticle transplacental transport mechanisms. Meanwhile, recent surface-functionalized nanoparticles for targeting the uterus and placenta are examined. Indeed, surface-functionalized nanoparticles could help prevent transplacental passage and promote placental-specific drug delivery, thereby enhancing efficacy and improving safety. We have achieved promising results in targeting the placenta via placental chondroitin sulfate A (plCSA), which is exclusively expressed in the placenta, using plCSA binding peptide (plCSA-BP)-decorated nanoparticles. Others have also focused on using placenta- and uterus-enriched molecules as targets to deliver therapeutics via surface-functionalized nanoparticles. Additionally, we propose that placenta-specific exosomes and surface-modified exosomes might be potential tools in the targeted therapy of pregnancy complications. Altogether, surface-functionalized nanoparticles have great potential value as clinical tools in the targeted therapy of pregnancy complications.

## 1. Introduction

It is estimated that more than 131 million births occur globally each year [1], of which over 10% of mothers and fetuses are affected by one or more pregnancy complications. These complications have two adverse aspects: One is fetal complications, such as intrauterine growth restriction (IUGR), pregnancy loss, and preterm birth and the other is maternal complications, such as preeclampsia and gestational diabetes mellitus (GDM) [2,3,4]. Although these pregnancy disorders are distressing and lead to substantial maternal and fetal or neonatal morbidity and mortality, highly effective strategies or medications are not available for pregnant women. Indeed, the number of clinical drugs in obstetrics is extremely limited, and only three new medications have been licensed over the past two decades [5,6]. In addition, most common medications are small molecules, which readily cross the placenta via passive diffusion and reach the fetus [7], leading to serious fetotoxicity, such as hemorrhage [8], teratogenicity [9], and adenocarcinoma [10]. Minimizing the exposure of the fetus to medication and reducing the adverse off-target effects in the mother are the primary challenges in developing modern drugs to treat pregnancy complications.

Nanomedicine is capable of resolving the aforementioned problems, because it employs nanotechnology to increase the safety and efficacy of medications by selectively delivering them to the intended site [11]. A poorly functioning placenta is the main cause of pregnancy complications [3,12]. Therefore, nanoparticles can be exploited to precisely control therapeutic drug delivery to the placenta with minimal risk of side effects in the fetus and mother. This offers a new opportunity for targeted treatment of pregnancy complications.

Although nanoparticles have been actively studied and clinically used for more than two decades in the field of oncology [13,14], their application to treat pregnancy complications is still in its infancy. The development of novel nanoparticle delivery systems for use in pregnancy requires an in-depth understanding of the structure of the placenta and the mechanism of placental nanoparticle transfer. Therefore, in this review, we first describe the human placental anatomy and then summarize currently known mechanisms of nanoparticle transfer across the placenta. At the same time, we will review nanoparticle-based therapeutics, focusing especially on improved surface-functionalized nanoparticle formulations for treating pregnancy complications.

## 2. Human Placental Anatomy

The placenta attaches to the uterine wall and is an important organ for maternal–fetal physiologic exchange and determines the extent of fetal exposure to maternally delivered nanoparticles. Placental shape and structure vary enormously between different species [15]. The human placenta is discoidal and hemochorial in structure and is filled with both fetal and maternal blood (Figure 1A,B). It is formed from both fetal (chorionic plate and chorionic villi) and maternal tissue (decidua basalis of the uterus). Decidua septa arising from the decidua basalis divide the placenta into separate functional units know as cotyledons [16,17]. Each cotyledon contains a villus tree in which the maternal and fetal tissues are separated by a placental barrier (Figure 1C). This barrier consists of fetal capillaries, endothelial cells, and an outer trophoblast, containing the villus stroma, the multinucleated syncytiotrophoblast, and the cytotrophoblasts [17]. In addition, the unique syncytiotrophoblast, which expresses a substantial number of specific transporters, is the rate-limiting layer for substances crossing the placenta [18,19,20]. Prior to the tenth week of pregnancy, maternal blood does not perfuse the placenta, and the placental barrier is not yet established; compounds that are present in the maternal blood may be delivered to the fetus via diffusion through extracellular fluid. As the placental barrier develops (from the 10th to the 12th week of pregnancy), drugs and other chemicals that have small molecular weights, are lipophilic, exhibit low protein binding, and have a low degree of ionization readily cross the placenta barrier [21].

## 3. Nanoparticle Transplacental Transport Mechanisms

The purpose of specifically controlling nanoparticles that reach the placenta is clear, yet our knowledge regarding nanoparticle transfer across the placental barrier is extremely limited, and several recent reviews have highlighted this as a new area in need of more research [22,23,24]. For nanoparticles to cross the placental barrier, they have to pass across two layers: (a) The syncytiotrophoblast and cytotrophoblasts layer and (b) the endothelial cell lining of fetal capillaries inside the villi. The possible transport routes for nanoparticles across the placenta are described below and summarized in Figure 2.

### 3.1. Paracellular Passage

From the results of in vitro drug-transport experiments, researchers have suggested that the placenta is a lipid-pore membrane [25]. Subsequently, through a solid body of evidence based on electron microscopy analysis, Kertschanska et al. demonstrated placental pores or channels that extend from the basal trophoblastic surface into the syncytiotrophoblast and range from 15 to 25 nm in diameter under normal intravascular pressure [26]. Furthermore, some studies have shown that the flexural channels in the placenta are continuous and pass from the fetal to the maternal side [27,28,29,30]. Therefore, placental channels would allow nanoparticles with a size under 25 nm to cross the placenta via passive diffusion transport (also known as paracellular passage). 

Paracellular passage is supported by an in vivo study showing that small quantum dots (QDs, < 25 nm) injected into pregnant mice cross the placenta more easily than larger QDs and the number of QDs transferred increases with increasing dosage [31]. However, data obtained from rodent experiments cannot be extrapolated to humans because the placenta is the most species-specific mammalian organ [32]. Ex vivo human placental perfusion provides a controlled and ethically accepted model that closely mimics the in vivo situation and can be used for studying transplacental transport of xenobiotics and particles. Using this model, studies have demonstrated that 25 and 50 nm silica particles [33] and 50 nm polystyrene nanoparticles can very quickly cross the placental barrier via simple passive diffusion [34]. Additionally, small dendrimer nanoparticles (5.6 nm) have been suggested to traverse the placenta through placental channels [35]. In contrast, bidirectional ex vivo perfusion studies have indicated that the transport of similar (50 nm) polystyrene nanoparticles in the fetal-to-maternal direction was significantly higher than transport in the reverse direction. The main mechanism of this translocation involves an active, energy-dependent pathway rather than passive diffusion [36]. Actually, the purported function of placental channels is to accommodate maternofetal blood flow. Under perfusion conditions, namely, hydrostatic and colloid osmotic pressure, the diameter of these channels can dilate to many times the normal size [37,38]. Enhanced uptake and passage of small gold nanoparticles was observed when placental structure and function were damaged by inflammation [39]. Hence, it is assumed that the conflicting findings may be ascribed to differences in perfusion pressure and placental conditions.

### 3.2. Transcellular Passage

Transcellular passage is an active process that includes both endocytosis and exocytosis, which involve complex vesicular systems and are strictly dependent on each other [40]. Endocytosis may be divided into phagocytosis (‘cellular eating’) and pinocytosis (‘cellular drinking’) [41,42]. Nanoparticle transport across the plasma membrane via transcellular passage may have multiple stages [40,43,44]. First, the nanoparticle is engulfed in membrane invaginations that form intracellular vesicles, also known as endosomes. Second, the vesicles deliver the nanoparticles to various specialized intracellular organelles, which enables sorting of vesicles toward different destinations. Finally, the nanoparticles loaded in the vesicles undergo intracellular trafficking to the opposite polar membrane and are excreted from the cells. The transport processes mentioned above are likely to be employed in mediating nanoparticle transport across the placenta.

#### 3.2.1. Endocytosis

Phagocytosis is the preferred route for uptake of nanoparticles larger than 500 nm and primarily occurs in a few types of professional phagocytes, including macrophages, monocytes, and dendritic cells, which are responsible for host defense and uptake of pathogens, dead cells, and debris [45,46]. In general, large nanoparticles are more efficiently taken up via phagocytosis. For example, for polystyrene nanoparticles in a size range of 1.0–2.0 μm, larger particles exhibit maximal uptake by mouse peritoneal macrophages [47]. Placental trophoblasts have also been suggested to display phagocytic activity, but to a much lesser degree [48,49,50]. While interesting, whether placental trophoblasts can take up nanoparticles via phagocytosis has not yet been demonstrated. 

Pinocytosis is the primary method for uptake of small nanoparticles and is classified as either clathrin-mediated endocytosis (CME) or clathrin-independent endocytosis [51]. Two major clathrin-independent endocytosis pathways are caveolae-mediated endocytosis and macropinocytosis. The placental syncytiotrophoblast displays many clathrin-coated regions between the microvilli, and coated vesicles, which are important participants in macropinocytosis, are abundant in the cytoplasm [52]. Hence, nanoparticles could be taken up by trophoblasts via pinocytosis. Evidence of this possibility came from studies showing that positively charged polymeric nanoparticles and gold nanoparticles were internalized by syncytiotrophoblasts via CME and caveolae-mediated endocytosis [53,54,55]. More recently, BeWo b30 placental barrier cells preincubated with a variety of different placental-transporter inhibitors also demonstrated that cells primarily internalized pullulan acetate nanoparticles through caveolae-mediated endocytosis and pinocytosis pathways [56]. Furthermore, macropinocytosis has been proposed for the uptake of small liposomes and nanoparticles (<60 nm) into placental trophoblasts and capillary endothelium [52,57].

#### 3.2.2. Exocytosis

Once nanoparticles are taken up by cells, there can be multiple possible routes, which can be used by the cells for nanoparticle removal [40,58,59]. While nanoparticle transport across the placental barrier may be also through the following exocytosis mechanisms: (a) Following endocytosis, nanoparticles can be internalized into early endosomes. An early endosome can mature into multivesicular bodies (MVBs), which will eventually fuse with the plasma membrane and release the nanoparticles outside the trophoblasts, and eventually, the nanoparticles will enter fetal circulation. (b) Some of the nanoparticles can exit the vesicles during endosomal maturation. Diffusion out of the trophoblasts can be the only way for such nanoparticles to exit the cells. (c) Early endosomes deliver nanoparticles to lysosomes, which can also undergo exocytosis and release their content into the villous stroma, followed by release into fetal capillaries. (d) Some cationic nanoparticles may be able to directly fuse with the negatively charged membrane of trophoblasts and move toward the basal membrane through simple diffusion, eventually diffusing into the fetal capillaries.

Among the macromolecules involved in the placental transcytosis mechanism, the most commonly studied is immunoglobulin G (IgG). Fc receptors (FcRs) on the apical membrane of the syncytiotrophoblast mediate IgG transport from maternal to fetal circulation [60,61]. IgG binding to FcR is pH dependent; the IgG–FcR complex has high affinity at pH 6.0 and is released at pH 7.4 [62,63]. After uptake in early endosomes, IgG bound to FcR can be transported to the syncytiotrophoblast basolateral plasma membrane and be released into the villous interstitium. Then, FcγR receptor located on the endothelium of the villous vasculature delivers IgG from the placenta to the fetal circulation via a receptor-mediated diffusion mechanism [64,65].

## 4. Surface-Functionalized Nanoparticles for Targeting the Uterus to Treat Pregnancy Complications

In order to carry out targeted therapy of pregnancy complications, several surface-functionalized nanoparticles that target the uterus and the placenta have been exploited (Table 1).

Oxytocin receptors (OTRs) are abundantly expressed in placental decidual tissues, derived from the uterus and uterine myometrium [66,67] but, in other tissues, including the brain, pituitary, and mammary tissue, OTR expression levels are low [68]. This indicates that the OTR is a candidate for the development of a targeted drug delivery system for the treatment of pregnancy complications. Based on this principle, immunoliposomes with the ability to target the uterus for the treatment of preterm labor via surface decoration with an anti-OTR antibody have been fabricated [69,70,71]. In vitro experiments showed that immunoliposomes loaded with contraction-blocking agents, including nifedipine, salbutamol, and rolipram, consistently abolished both human and mouse uterine contractility [69]. The results of in vivo experiments with pregnant mice showed that immunoliposomes efficiently delivered indomethacin to the uterus and reduced the rates of preterm birth compared with untargeted liposomes. Additionally, no evidence of transplacental passage of the immunoliposomes to the fetus was observed [69]. 

Since the 1980s, immunoliposomes have been studied as a therapeutic tool for cancer and other diseases [88,89,90]. However, immunoliposomes have not yet entered clinical application. The highest risk associated with immunoliposomes is acute immune reactions, for example, anaphylaxis, anaphylactoid reactions, and serum sickness [91]. Pregnancy complications, such as preeclampsia and preterm birth, are proinflammatory in nature and characterized by immune cell activation [92]. Therefore, although fetal exposure to the drugs is minimized, administration of immunoliposomes may potentially aggravate maternal symptoms.

Similarly, liposomes conjugated with atosiban, a clinically available OTR antagonist [93], were also designed to target the uterus for preterm labor management in pregnant mice [72]. The atosiban-modified liposomes significantly increased the indomethacin concentration in the uterus, successfully prevented indomethacin passage to the fetus, and thus reduced the rate of preterm birth. At therapeutic concentrations, atosiban has been demonstrated to be safe [93], and the total injected dose of the OTR antagonist in this study was several times lower than the minimal therapeutic dose. Interestingly, in vitro studies comparing OTR antibody- and antagonist-decorated liposomes showed that these two drug delivery systems have similar OTR binding ability and cellular uptake through both clathrin- and caveolin-mediated mechanisms [71]. Altogether, OTR is a novel target for uterine drug delivery during pregnancy, and OTR-targeted nanoparticles may provide an effective tool for targeting the uterus to treat pregnancy complications.

## 5. Surface-Functionalized Nanoparticles for Targeting the Placenta to Treat Pregnancy Complications

### 5.1. EGFR Antibody

Several research groups across the globe are now devoted to developing placenta-targeted nanoparticles to treat pregnancy complications. Because epidermal growth factor receptor (EGFR) expression is highest in placental trophoblasts compared with all other nonmalignant normal human tissues, Kaitu’u-Lino et al. developed doxorubicin-loaded nanocells to treat ectopic pregnancy by coating their surface with EGFR-targeting bispecific antibodies [73]. The EGFR-targeted nanocells were readily taken up by human placental explants and significantly increased placental choriocarcinoma cell death in vitro. In a mouse subcutaneous tumor model, EGFR-targeted nanocells were found to successfully inhibit tumor cell growth and reduce tumor volume compared with nontargeted nanocells. This study was the first one to address the viability of surface-functionalized nanoparticle-mediated drug therapy in both in vitro and in vivo settings. However, toxicity and clearance data were not presented. It should be determined to what extent the EGFR-targeted nanocells will spare normal nonplacental tissues, such as the kidney and liver, from exposure to the drug in pregnant women.

Some investigators have observed that the use of full-length anti-EGFR antibodies for the treatment of diseases, especially cancer [94,95,96], has limitations due to the large size of the molecules and their long blood circulation half-life [80]. In addition, the specificity of EGFR-targeted nanoparticles may be affected by interaction of the whole antibody with Fc receptors on normal tissues [81]. To overcome these problems, single-chain antibody fragments against EGFR (ScFvEGFR) that consist of the heavy and light variable chains but lack the Fc region have been developed. ScFvEGFR-conjugated nanoparticles have exhibited the same affinity and specificity for the target antigen as the parent full-length antibodies [80,82]. While interesting, successful targeting to the EGFR-overexpressing placenta has not yet been demonstrated using ScFvEGFR, and its potential applicability remains to be determined.

### 5.2. Tumor Homing Peptides

The tumor homing peptides CGKRK and iRGD, which selectively target integrins on the placenta and uterine spiral arteries, have also been used for placenta-specific drug delivery [74]. Lynda Harris and team demonstrated that liposomes decorated with the tumor homing peptides were able to successfully target the placental trophoblasts and the endothelium of established uterine spiral arteries. When the targeted liposomes were loaded with insulin-like growth factor 2 and administered to a fetal growth restriction mouse model, significantly enhanced fetal and placental growth was observed, and there was growth factor transfer to the fetus. Additionally, this group developed homing-peptide miRNA inhibitor conjugates [97]. In vivo experiments showed that the conjugates significantly increased fetal and placental weights compared with controls. This team also identified other novel uteroplacental-targeted peptides using a phage screening method [75]. Similarly, targeted peptide-decorated liposomes specifically bound to the spiral arteries and placental labyrinth in pregnant mice. The targeted liposomes efficiently delivered a NO donor (SE175) to the placenta and significantly increased fetal weight and mean spiral artery diameter in pregnant endothelial nitric oxide synthase (eNOS) knockout mice. This group has developed a number of placental homing peptide-decorated liposomes that show great promise as potential therapeutics for fetal growth restriction.

### 5.3. plCSA-BP

*Plasmodium falciparum* has evolved a protein, VAR2CSA, that mediates the binding of infected erythrocytes to placental chondroitin sulfate A (plCSA) on the surface of syncytiotrophoblasts, leading to accumulation in the placenta but not in other tissues [98,99,100]. Meanwhile, a peptide from the minimal plCSA-binding region of VAR2CSA was identified through a phage screening approach [101], and we named the synthetic peptide placental CSA binding peptide (plCSA-BP). Through histological analysis, we demonstrated that plCSA-BP could specifically bind to a distinct placental-type CSA expressed on trophoblasts in human and mouse placentas. Notably, plCSA-BP has the ability to specifically bind to trophoblasts throughout gestation in the mouse [76]. We also synthesized plCSA-BP-conjugated nanoparticles (plCSA-NPs) [77] and demonstrated successful delivery of methotrexate specifically to placental trophoblasts via plCSA-NPs [76]. Importantly, plCSA-NPs had no adverse effects on maternal tissues and did not detectably cross the placenta. In addition, nanoparticles decorated with plCSA-BP were efficiently taken up into lysosomes by choriocarcinoma JEG3 cells, and in vivo experiments showed that targeted nanoparticles loaded with doxorubicin significantly inhibited tumor growth and metastasis relative to controls [78]. Meanwhile, we developed the first three-complementary method that utilizes in vivo imaging, high-frequency ultrasound, and high-performance liquid chromatography (HPLC) as a novel tool to evaluate the effectiveness and safety of placenta-targeted drug delivery systems [79]. Altogether, plCSA is a specific molecule for targeting the placenta, and plCSA-BP-decorated nanoparticles could be an effective tool for delivering large amounts of therapeutic drugs to treat pregnancy complications.

## 6. Untapped Potential of Placenta-Targeted Tools for Treatment of Pregnancy Complications

In multicellular organisms, cells have the ability to communicate with each other in order to proliferate and maintain homeostasis. In recent years, an extensive number of studies have demonstrated that exosomes serve important roles in cell communication with neighboring and distant cells [102,103]. Exosomes are lipid bilayer nanovesicles with a distinct size (between 30 and 150 nm) and a buoyant density of 1.12–1.19 g/mL. They are formed when MVBs undergo reverse budding and are released when MVBs fuse with the cytoplasmic membrane [104]. 

Exosomes are continuously shed from trophoblast cells into maternal circulation throughout gestation. Recently, it was demonstrated that chorionic villous trophoblasts (CVTs) are capable of modulating the activity of extravillous trophoblasts (EVTs) via exosomal miRNA [105,106,107]. Moreover, the number of exosomes produced during gestation is higher when pregnancy complications occur, such as preeclampsia and gestational diabetes mellitus (GDM), than in normal pregnancies [83]. Specifically, the number of placenta-derived exosomes is massive, and they can be taken up by trophoblasts. The level and content of placenta-specific exosomes may be a useful tool for the delivery of payloads to the placenta. Exosomes loaded with doxorubicin and siRNA have been reported as a feasible approach with great potential value for clinical applications [84,85]. Exosomes as drug delivery vehicles have multiple advantages. First, exosomes can be derived from a patient’s own cells, and thus, they may be less immunogenic and toxic than artificial nanoparticles [108]. Second, exosomes have lipid bilayers, which may directly fuse with target cell membranes, thereby increasing cellular internalization of loaded drugs [109]. Third, the naturally small size of exosomes allows them to avoid phagocytosis by the mononuclear phagocyte system and facilitates their uptake in target tissues [85]. In addition, exosomes have also been proposed to specifically recognize their target cells, a feature that would reduce off-target effects [110,111]. Hence, placenta-specific exosomes may deliver payloads to the placenta by directly targeting trophoblasts. If not, placenta-specific exosome surfaces can be modified with targeting peptides or antibodies by manipulating trophoblasts, either through genetic or metabolic engineering or by introducing exogenous material that is subsequently incorporated into secreted exosomes (Figure 3). For example, the targeting peptides RVG (rabies viral glycoprotein peptide) and iRGD were successfully inserted into exosomes from immature dendritic cells using a genetic method [86,87]. Overall, re-engineering placenta-specific exosomes is a very promising approach for the development of bio-inspired, targeted drug delivery systems to treat pregnancy complications. 

The untapped potential of certain placenta-enriched molecules, including placental growth factor (PlGF), pregnancy-associated plasma protein-A (PAPP-A), and soluble FMS-like tyrosine kinase 1 (sFlt-1), have been highlighted previously [112]. Targeting these circulating placental proteins might be another method to treat pregnancy complications. Recently, direct targeting of sFlt-1 has been reported via siRNA therapies administered in a baboon preeclampsia model [113]. Meanwhile, a better understanding of placenta biology and the pathogenesis of pregnancy complications may yield more placenta-targeted tools.

## 7. Conclusions and Perspectives

Nanomedicines for the treatment of a variety of diseases are already on the market, and the number of nanomedicine products is growing rapidly with clinical development [114]. In line with that, nanomedicine for the targeted therapy of pregnancy complications is showing great promise; however, more studies are needed. A deep understanding of the translocation and effects of nanoparticles in the human placenta is necessary for the development of novel nanoparticle platforms to treat pregnancy complications while reducing fetal exposure. A key consideration when studying nanoparticle transfer across the placenta is that it may involve multiple mechanisms. To give one example, polystyrene nanoparticles with a size of approximately 50 nm may cross the placenta via diffusion and the transtrophoblastic channel system, but those with sizes up to 120 nm may cross the placenta through caveolin-coated vesicles [34]. Hence, nanoparticle transfer across the placental barrier is dependent on particle characteristics and functionalization [115]. Additionally, restricting nanoparticle retention in the placenta by inhibiting the placenta transport system is difficult. Surface-functionalized nanoparticles are a good choice for promoting nanoparticle uptake by the placenta. Meanwhile, one of the key problems is finding a target for binding that is exclusively placental. In our study, plCSA-BP achieved that goal and it is an efficient tool that can selectively bind placental trophoblasts. The clinical need for new methods to deliver drugs in pregnancy is substantial, and therefore, exploring more tools is a timely endeavor. Placenta-specific exosomes may have placenta homing characteristics, and we propose that re-engineered placenta-specific exosomes can be used directly as a tool for the targeted delivery of therapeutics to the placenta. If not, surface-functionalized exosomes constitute a very promising approach for development of bio-inspired, targeted drug delivery systems to treat pregnancy complications. In summary, the use of nanomedicine in pregnancy is a new frontier in perinatal therapeutics, and surface-functionalized nanoparticles could be efficient tools in the targeted therapy of pregnancy complications.

## Figures and Tables

**Figure 1 ijms-20-03642-f001:**
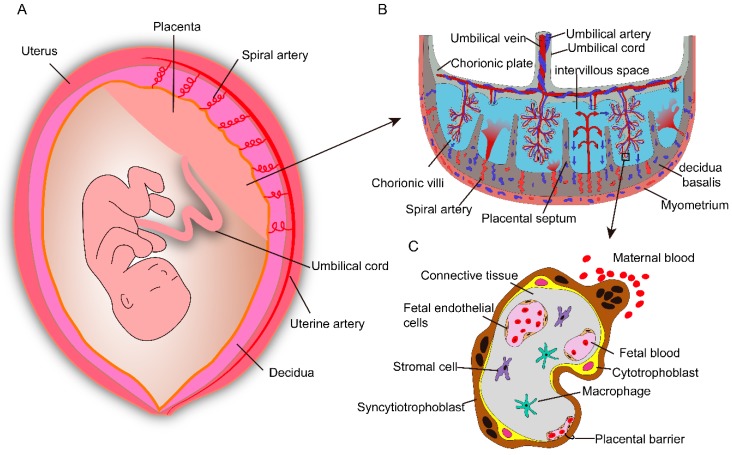
Schematic depiction of anatomical structures of the human placenta. (**A**) The placenta is an important organ for maternal–fetal physiologic exchange and attaches to the uterine wall. (**B**) Anatomical structure and composition of the human placenta. The umbilical vein transport oxygen- and nutrient-rich blood from the placenta to fetus, while two umbilical arteries carry waste products from the fetus to the placenta. The intervillous space is filled with maternal blood, which enters through remodeled spiral arteries. (**C**) The major cell types and the placental barrier, it consists of syncytiotrophoblasts, cytotrophoblasts, and fetal endothelial cells.

**Figure 2 ijms-20-03642-f002:**
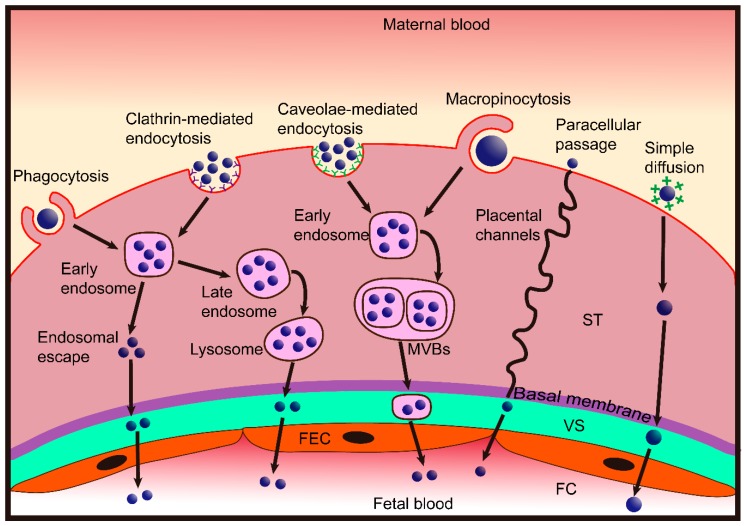
Scheme showing the nanoparticle transplacental transport mechanisms. Nanoparticles cross the placental barrier by paracellular passage and transcellular passage. Very small nanoparticles can penetrate syncytiotrophoblasts (ST) via placental channels and thereby enter the villous stroma (VS). Diffusion may then occur through the fetal endothelial cells (FEC) into the lumen of the fetal capillaries (FC). The transcellular passage consists of endocytosis and exocytosis. Nanoparticles may be taken up by syncytiotrophoblasts via phagocytosis, clathrin-mediated endocytosis, caveolae-mediated endocytosis and macropinocytosis, then they may be exocytosed through the endosomal escape pathway, lysosomal secretion, and multivesicular bodies (MVBs)-related secretion. After entering the VS, nanoparticles may cross the FEC by the same pathway and eventually enter into the fetal blood. Some cationic nanoparticles may be able to directly fuse the negatively charged trophoblast cell membrane and move toward the basal membrane by simple diffusion, thereby, diffusing through the FEC into fetal blood.

**Figure 3 ijms-20-03642-f003:**
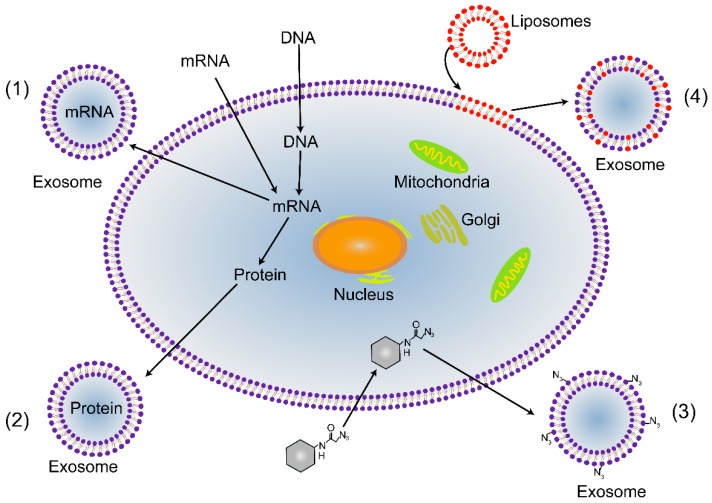
Strategies for re-engineering placenta-targeted exosomes. (1) Genetic engineering can be used to introduce mRNA and DNA into trophoblasts. It can be incorporated into exosomes to promote gene expression or regulate transcription in recipient trophoblasts. (2) Transgenic proteins can be packaged into exosomes, for example, as targeting moieties and fluorescent reporters. (3) Metabolic engineering, in which metabolite analogs are packaged into trophoblast biosynthesis. This method can be used to introduce functional groups; for instance, azides can be attached to the surface of exosomes, which allows subsequent bio-orthogonal reactions to be performed. (4) Material can be incorporated into exosomes via liposomes that fuse with cytoplasmic membranes.

**Table 1 ijms-20-03642-t001:** Surface-functionalized nanoparticles as tools in targeted therapy of pregnancy complications.

Decorated Ligands/Peptides	Nanoparticles Type	Conjugated Method	Targeted Organ	Pregnancy Complications	Ref.
OTR-antibody	Immunoliposomes	Michael-type addition reaction	Uterus	Preterm birth	[69,70,71]
Atosiban	Liposomes	Post-insertion technique	Uterus	Preterm birth	[71,72]
EGFR antibody	Nanocells	Bispecific antibodies	Placenta	Ectopic pregnancies	[73]
Tumor homing peptides	Liposomes	Michael-type addition reaction	Placenta	Fetal growth restriction	[74]
Uteroplacental-targeted peptide	Liposomes	Michael-type addition reaction	Placenta	Fetal growth restriction	[75]
plCSA-BP	Lipid-polymer nanoparticles	EDC/NHS	Placenta	Normal pregnancy, Choriocarcinoma	[76,77,78,79]
ScFvEGFR antibody	Untapped	EDC/NHS	Placenta	Untapped	[80,81,82]
Untapped	Placenta-derived exosomes	Untapped	Placenta	Untapped	[83,84,85,86,87]

OTR—oxytocin receptor; EGFR—epidermal growth factor receptor; plCSA-BP—placental chondroitin sulfate A-binding receptor; ScFvEGFR—single-chain antibody fragments against EGFR.

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
