# Peer review of "Surface-Functionalized Nanoparticles as Efficient Tools in Targeted Therapy of Pregnancy Complications"

_ijms, 2019, doi:10.3390/ijms20153642_

Round 1

Reviewer 1 Report

Dear authors

Let me congratulate, first, for the work. Very interessting review of a very difficult topic.I agree totally with the wy you've done, but I have some concerns and suggestions to you:

- Line 37 to 39, please consider to revise the text.

- Line 91, as well, consider possible revission, specially after the citation numbers

- Line 113, show, should be shows, in my opinion

- Line 119, you talk about the perfusion model with no figures or complemantary explanations about it, what makes some confusion for readers. After in the decurs of the text, you talk, frequently about the diffusion through different memebranes and tissues, why not to make a shor paragraph showing the different possible mechanisms ? Or make it in Fig. 2

-Line 142, "larger" instead "large"

. Line 155, please consider the time of the verb used

. As well in Line 164

. Line 187, consider to include commma, after the word tissue.

- Line 228-229 the phrase, in my opinion, has no sense, when trying to explain the changes in advanced tumors,

- Line 272-273, consider to revise the part referred to the techniques that can be applied

In general, I consider that the article would need some more Figures. Some of the explanations about mechanisms and phenomenology, that, in my opinion, are specially interessting. When talking about exosomes mechanisms and Lyosomes

- The explanation about the mechanisms of "decorated nanoparticles" will be more interessting with the help of figures or schemes of about how nanoparticles have been "decorated"

Excellent work

Author Response

Response to Reviewer 1 Comments

Point 1:Line 37 to 39, please consider to revise the text.

Response 1: We have completely re-written Line 37 to 39 as suggested (Line 38-42).

Point 2: Line 91, as well, consider possible revission, specially after the citation numbers

Response 2: As suggested, we have re-written this sentence in the revised manuscript (Line 93-94).

Point 3: Line 113, show, should be shows, in my opinion

Response 3: Thank you very much. We corrected it in the revised manuscript.

Point 4: Line 119, you talk about the perfusion model with no figures or complemantary explanations about it, what makes some confusion for readers. After in the decurs of the text, you talk, frequently about the diffusion through different memebranes and tissues, why not to make a shor paragraph showing the different possible mechanisms ? Or make it in Fig. 2

Response 4: Thank you for your suggestion. We have described the perfusion model in the revised manuscript (Line 123-127). Additionally, we have revised the legend of figure 2, the possible different mechanisms of nanoparticle diffusion through different membranes of placental barrier has been detailed described (Figure 2 Legend).

Point 5: Line 142, "larger" instead "large"

Response 5: Thank you. We have corrected it in the revised manuscript.

Point 6: Line 155, please consider the time of the verb used

Response 6: We have corrected it. Meantime, we have carefully checked grammars in the revised manuscript.

Point 7: As well in Line 164

Response 7: We have corrected it in the revised manuscript.

Point 8: Line 187, consider to include commma, after the word tissue.

Response 8: Thank you. These errors have been corrected in the revised manuscript.

Point 9: Line 228-229 the phrase, in my opinion, has no sense, when trying to explain the changes in advanced tumors,

Response 9: Thank you very much for your instructive suggestions. We have deleted the phrase.

Point 10: Line 272-273, consider to revise the part referred to the techniques that can be applied

Response 10: Thank you for your suggestions. We have revised this part (Line 290).

Point 11: In general, I consider that the article would need some more Figures. Some of the explanations about mechanisms and phenomenology, that, in my opinion, are specially interessting. When talking about exosomes mechanisms and Lyosomes

Response 11: Thank you for your suggestion. We have added Table 1 and Figure 3 in the revised manuscript.

Point 12: The explanation about the mechanisms of "decorated nanoparticles" will be more interessting with the help of figures or schemes of about how nanoparticles have been "decorated"

Response 12: As suggested, we have added conjugating method in Table1, and added Figure 3 to describe the strategies for re-engineering placenta-targeted exosomes. Thank you again.

Reviewer 2 Report

The manuscript describes surface-functionalized nanoparticle formulations for treating pregnancy complications. My only constructive comment is to add Tables and representative images in Chapter 4,5 and 6 with related references to improve the readability of readers.

Author Response

Point 1: The manuscript describes surface-functionalized nanoparticle formulations for treating pregnancy complications. My only constructive comment is to add Tables and representative images in Chapter 4,5 and 6 with related references to improve the readability of readers.

Response 1: Thank you very much for your instructive suggestions. We have added Table1 to summarize chapter 4-6, and added Figure 3 to describe the strategies for re-engineering placenta-targeted exosomes in the revised manuscript.

Round 2

Reviewer 2 Report

According to the reviewer's suggestion, the paper has undergone changes and is improved compared to the first manuscript. Most responses to the reviewer's questions were well addressed and reasonable in the revised manuscript. Therefore, I recommend that the revised manuscript is suitable for the publication in IJMS.